# Single Dose of Attenuated Vaccinia Viruses Expressing H5 Hemagglutinin Affords Rapid and Long-Term Protection Against Lethal Infection with Highly Pathogenic Avian Influenza A H5N1 Virus in Mice and Monkeys

**DOI:** 10.3390/vaccines13010074

**Published:** 2025-01-15

**Authors:** Fumihiko Yasui, Keisuke Munekata, Tomoko Fujiyuki, Takeshi Kuraishi, Kenzaburo Yamaji, Tomoko Honda, Sumiko Gomi, Misako Yoneda, Takahiro Sanada, Koji Ishii, Yoshihiro Sakoda, Hiroshi Kida, Shosaku Hattori, Chieko Kai, Michinori Kohara

**Affiliations:** 1Department of Microbiology and Cell Biology, Tokyo Metropolitan Institute of Medical Science, 2-1-6, Kamikitazawa, Setagaya-ku, Tokyo 156-8506, Japan; 2Infectious Disease Control Science, Institute of Industrial Science, The University of Tokyo, 4-6-1, Komaba, Meguro-ku, Tokyo 153-8505, Japan; 3Animal Laboratory of Injurious Animals, The Institute of Medical Science, The University of Tokyo, 802, Tean Sude, Setouchi-cho, Oshima-gun, Kagoshima 894-1531, Japan; 4Center for Quality Management Systems, National Institute of Infectious Diseases, 4-7-1, Gakuen, Musashi-murayama, Tokyo 208-0011, Japan; 5Laboratory of Microbiology, Faculty of Veterinary Medicine, Hokkaido University, Kita 18 Nishi 9, Kita-ku, Sapporo 060-0818, Japan; 6Institute for Vaccine Research and Development (HU-IVReD), Hokkaido University, Sapporo 001-0021, Japan; 7International Institute for Zoonosis Control, Hokkaido University, Sapporo 001-0020, Japan

**Keywords:** vaccinia virus vector, cross-clade protection, hemagglutinin protein, rapid protection, long-term protection

## Abstract

Background/Objectives: In preparation for a potential pandemic caused by the H5N1 highly pathogenic avian influenza (HPAI) virus, pre-pandemic vaccines against several viral clades have been developed and stocked worldwide. Although these vaccines are well tolerated, their immunogenicity and cross-reactivity with viruses of different clades can be improved. Methods: To address this aspect, we generated recombinant influenza vaccines against H5-subtype viruses using two different strains of highly attenuated vaccinia virus (VACV) vectors. Results: rLC16m8-mcl2.2 hemagglutinin (HA) and rLC16m8-mcl2.3.4 HA consisted of a recombinant LC16m8 vector encoding the HA protein from clade 2.2 or clade 2.3.4 viruses (respectively); rDIs-mcl2.2 HA consisted of a recombinant DIs vector encoding the HA protein from clade 2.2. A single dose of rLC16m8-mcl2.2 HA showed rapid (1 week after vaccination) and long-term protection (20 months post-vaccination) in mice against the HPAI H5N1 virus. Moreover, cynomolgus macaques immunized with rLC16m8-mcl2.2 HA exhibited long-term protection when challenged with a heterologous clade of the HPAI H5N1 virus. Although the DIs strain is unable to grow in most mammalian cells, rDIs-mcl2.2 HA also showed rapid and long-lasting effects against HPAI H5N1 virus infection. Notably, the protective efficacy of rDIs-mcl2.2 HA was comparable to that of rLC16m8-mcl2.2 HA. Furthermore, these vaccines protected animals previously immunized with VACVs from a lethal challenge with the HPAI H5N1 virus. Conclusions: These results suggest that both rLC16m8-mcl2.2 HA and rDIs-mcl2.2 HA are effective in preventing HPAI H5N1 virus infection, and rDIs-mcl2.2 HA is a promising vaccine candidate against H5 HA-subtype viruses.

## 1. Introduction

Some H5 and H7 subtypes of avian influenza viruses are known to become highly pathogenic avian influenza (HPAI) viruses under natural conditions. HPAI viruses have caused outbreaks of acute systemic diseases in terrestrial birds, particularly poultry, with a mortality rate that often approaches 100%. The HPAI H5N1 virus has occasionally infected humans with a high mortality rate. To date, 904 laboratory-confirmed human cases of HPAI H5N1 virus infection have been reported in 17 countries, with 464 fatal cases [1]. The impact of the HPAI H5N1 virus on animal and public health has led to the prediction that the H5-subtype virus could potentially trigger the next pandemic [2,3,4,5], since this hemagglutinin (HA)-subtype virus is distinct from viruses circulating in the human population.

Vaccination is the most effective strategy for preventing infectious diseases, and the development of effective vaccines against the HPAI virus is important for pandemic preparedness. Although inactivated vaccines are safe, their immunogenicity is low. Therefore, numerous recombinant viral vectors have been used to develop novel vaccine candidates targeting a wide range of infectious diseases, including HPAI H5N1 [6,7,8,9,10]. Among them, highly attenuated vaccinia viruses (VACVs) have garnered attention as promising vectors because of their safety and immunogenicity in humans. LC16m8 is a VACV strain and the sole smallpox vaccine licensed in Japan. A clinical evaluation of LC16m8 was conducted in over 50,000 children in the 1970s, and the results showed no serious adverse events, including encephalitis, after vaccination [11,12]. Moreover, LC16m8 has been used to inoculate army personnel in Japan and the USA [13,14,15]. In previous studies, we have demonstrated the protective effects of recombinant LC16m8 strains expressing the gene encoding either the spike protein of SARS-CoV or the hemagglutinin protein of the HPAI H5N1 virus [16,17]. The DIs strain is derived from the DIE strain of VACV via extensive serial passaging in chicken embryo fibroblasts (CEFs) [18] and exhibits a restricted host range because of a large-scale deletion of the genome (approximately 15.4 kb, or 8% of the parental genome), which has resulted in the loss of its the ability to replicate in most mammalian cells [19]. The DIs strain is also a useful viral vector for developing recombinant vaccines against viral diseases [20]. Given that DIs do not replicate in various mammalian cells, DIs-based recombinant vaccines are likely safer for immunodeficient and immunocompromised populations. Hence, these two attenuated VACV strains are promising vectors for the development of novel vaccines.

In this study, we aimed to investigate the rapid and long-term protective efficacy of HPAI H5N1 virus vaccine candidates based on two different types of highly attenuated VACV vectors (DIs and LC16m8 strains). The objective was to test vaccine candidates against lethal challenges with heterologous HPAI H5N1 virus in mouse and non-human primate models.

## 2. Material and Methods

### 2.1. Ethics Statement

All experiments involving primary rabbit kidney (pRK) cells, chicken embryonic fibroblasts (CEFs), and mice were approved by the Tokyo Metropolitan Institute of Medical Science Animal Experiment Committee (approval number: 22-2, 11-081, 1282, 13035, 17066, 18050) and performed in accordance with the animal experimentation guidelines of the Tokyo Metropolitan Institute of Medical Science. All macaque experiments were approved by the Animal Experimental Committee of the Institute of Medical Science at the University of Tokyo (approval number: PH11-24) and performed in accordance with the Regulations for Animal Care and Use of the University of Tokyo.

### 2.2. Cells

RK13 cells were maintained in Eagle’s Minimum Essential Medium (MEM; Nissui Pharmaceutical Co., Ltd., Tokyo, Japan) containing 5% fetal bovine serum (FBS). Madin-Darby canine kidney (MDCK) cells were maintained in Dulbecco’s Modified Eagle’s medium (DMEM; Nissui Pharmaceutical Co., Ltd., Tokyo, Japan) supplemented with 10% FBS. pRK cells were prepared to construct and propagate LC16m8-derived recombinant VACVs carrying the HA protein-encoding gene of the H5N1 influenza virus, as previously described [17]. Seven-day-old chicken embryos were collected in Hanks’ Balanced Salt Solution [HBSS (-)] containing 50 U/mL penicillin, 50 μg/mL streptomycin, and 0.1% glucose. After removing the eyes, brain, beaks, wings, and feet from each embryo, the rest of the body was minced with scissors and digested using TrypLE Select Select (Thermo Fisher Scientific, Waltham, MA, USA). The cells were grown in DMEM containing 10% newborn calf serum and tryptose phosphate broth.

### 2.3. Viruses

The H5N1 HPAI virus A/whooper swan/Hokkaido/1/2008 and A/whooper swan/Mongolia/3/2005 strains were grown in embryonated eggs. HPAI virus A/Vietnam/UT3040/2004, isolated from a human patient in Vietnam [21], was kindly provided by Dr. Yoshihiro Kawaoka (University of Tokyo, Tokyo, Japan). The A/Vietnam/UT3040/2004 strain was propagated in MDCK cells grown in MEM containing 1% bovine serum albumin (BSA) and 10 μg/mL acetyl-trypsin. All procedures using the H5N1 HPAI were performed in biosafety level 3 facilities by personnel wearing powered air-purifying respirators (Shigematsu Co., Ltd., Tokyo, Japan). The VACV strain LC16m8 was kindly provided by KAKETSUKEN Co., Ltd. (Kumamoto, Japan). LC16m8 was propagated in pRK cells. The VACV strain DIs was propagated in pCEFs.

### 2.4. Generation of Recombinant Vaccinia Viruses Encoding H5 HA

Synthetic DNA encoding the modified HA of A/bar-headed goose/Qinghai Lake/1A/05 was purchased from Sloning Bio Technology (Puchheim, Germany). Specifically, the nucleotide sequence was altered such that the highly pathogenic HA cleavage site (QGERRRKKRGLF) of the encoded protein was mutated to the protein sequence (QIETRGLF) of the HA of a low-pathogenicity virus. The modified *HA* gene was used to replace the *HA* gene region of the LC16m8 genome (rLC16m8-mcl2.2 HA) via homologous recombination, as previously described [17]. rLC16m8-mcl2.3.4 HA, which carries DNA encoding a multibasic-site-deleted version of the HA protein of the H5N1 HPAI virus A/Anhui/01/2005 (H5N1)-PR8-IBCDC-RG5 influenza virus [clade 2.3.4] has been described previously [17]. Recombinant LC16m8 with only promoter sequences, but not an H5 HA-encoding gene (rLC16m8-empty), which has also been described in a previous study [17], was used as a control.

Recombinant DIs encoding the HA protein of H5N1 clade 2.2 (rDIs-mcl2.2 HA) were was generated as described previously [19]. Synthetic DNA (described above) was amplified by PCR using primers that incorporated flanking restriction enzyme sites (HindIII) and inserted by cloning into a plasmid (pUC/DIs) designed for recombination with DIs. The resulting pUC/DIs-mcl2.2 plasmid was linearized by digestion with restriction enzymes, and pCEFs that had been infected with DIs at a multiplicity of infection (MOI) of 10 for 1 h were transfected with the plasmid. After 24 h, the virus–cell mixture was harvested by scraping the cell layer, and the resulting suspension was frozen at −80 °C until further use. rDIs-mcl2.2 HA was purified and propagated as described previously [17]. The DIs were used as a control.

### 2.5. Mouse Study

Female BALB/c mice were purchased from SLC (Shizuoka, Japan), provided ad libitum access to food and water, and maintained under conditions involving a 12 h light/12 h dark cycle. For vaccination, 8-week-old animals were anesthetized by intraperitoneal administration of 0.15 mL/mouse of a ketamine–xylazine mixture and intradermally (on their dorsum) administered either 1 × 10^7^ plaque-forming units (PFU) of rLC16m8-H5 HA (clade 2.2 or clade 2.3.4) or rDIs-H5 HA (clade 2.2). After vaccination, the mice were inoculated intranasally with 1 × 10^4^ PFU of the H5N1 A/whooper swan/Hokkaido/1/2008 strain or 150 PFU of the A/Vietnam/UT3040/2004 strain. Clinical signs were monitored for 9, 12, or 14 days post-infection (dpi), and the mice were euthanized at 1, 3, 6, 9, 12, or 14 dpi to collect organ samples and sera. Body weight was monitored daily, and mice that lost 30% or more of their initial body weight were recorded as dead or euthanized. In the pre-immunization VACV study, BALB/c mice were inoculated intradermally with 1 × 10^7^ PFUs of LC16m8 or rLC16m8-empty. After 4 weeks, the rLC16m8-empty-immunized mice were intradermally inoculated with either 1 × 10^7^ PFU of rLC16m8-mcl2.2 HA or rLC16m8-empty and inoculated intranasally with 1 × 10^4^ PFU of the H5N1 A/whooper swan/Hokkaido/1/2008 strain. In contrast, sera were collected 21 weeks after LC16m8 inoculation from LC16m8-immunized mice and used to determine the neutralization titer against LC16m8 using the plaque assay, as described below. LC16m8-immunized mice were divided into four subgroups, in which the neutralization titer against VACV was similar within each subgroup (Appendix A), and inoculated intradermally with rDIs-mcl2.2 HA (1 × 10^7^, 3 × 10^7^, or 1 × 10^8^ PFU/mouse) or DIs (1 × 10^8^ PFU/mouse). All mice were infected with 1 × 10^4^ PFUs of the H5N1 A/whooper swan/Hokkaido/1/2008 strain 5 weeks after vaccination.

### 2.6. Macaque Study

Three- to five-year-old female cynomolgus macaques (2.5–3.5 kg each) obtained from the Philippines (Ina Research Inc., Ina, Japan) were used in this study. Prior to use, the macaques were confirmed to be free from hepatitis B virus, hepatitis E virus, *Mycobacterium tuberculosis*, *Shigella* spp., *Salmonella* spp., and *Entamoeba histolytica*. The animals were fed CMK-2 (CLEA Japan, Inc., Tokyo, Japan) once a day. Drinking water was provided ad libitum. Animals were individually housed under controlled conditions of humidity (40 ± 5%), temperature (25 ± 1 °C), and light (12 h light/12 h dark cycle, lights on at 8:00 a.m.). All procedures were performed under the influence of ketamine and xylazine anesthesia, and all efforts were made to minimize suffering. In the text and figures, individual macaques have been distinguished by their identification numbers. The absence of antibodies against the anti-influenza A virus in the sera of these animals was confirmed using an enzyme-linked immunosorbent assay (ELISA), which was used to examine the presence of antibodies against purified HA protein.

The macaques were inoculated intradermally (in both upper arms using 30G needles) with 1 × 10^7^ PFUs (5 × 10^6^ PFU/arm × 2) of rLC16m8-mcl2.2 HA or rLC16m8-empty. Macaques inoculated with rLC16m8-empty were used as controls. Sera were collected monthly for use in the ELISA. Transplants of telemetry (TA10CTA-D70, Data Sciences International) were performed one to two months before the viral challenge, as described previously [22], and were used to monitor the body temperatures of the macaques. One year after vaccination, the macaques were inoculated via the intratracheal, intranasal, and tonsillar routes with 3 × 10^6^ PFUs of H5N1 A/whooper swan/Hokkaido/1/08. Clinical signs were monitored for 7 days post-challenge, and the macaques were euthanized under anesthesia at the end of the experimental period.

### 2.7. ELISA

The production of immunoglobulin G (IgG), IgG1, and IgG2a specific to H5 clade 2.2. HA protein in the sera of mice and IgG levels specific to the H5 HA protein in the sera of cynomolgus macaques was measured using an ELISA. Recombinant H5 HA clade 2.2 protein tagged with a 6× His tag was produced by infecting cells with recombinant VACV LC16mO, which was generated to carry DNA encoding the modified HA protein tagged with a C-terminal 6× His tag (rLC16mO-mcl2.2 HA-His), as described previously [16]. The His-tagged HA protein was purified from the cell lysate using a Ni-Sepharose 6 Fast Flow resin column (GE Healthcare) and dialyzed against phosphate-buffered saline [PBS(-)]. ELISA plates were coated with 0.2 μg/mL of H5 HA solution at 4 °C overnight and then blocked with a blocking buffer [1% BSA/0.5 mM EDTA/0.5% Tween 20/PBS(-)] at room temperature for 2 h. The sera of mice and macaques were diluted 1000-fold with a dilution buffer [1% BSA/0.5 mM EDTA/0.5% Tween 20/PBS(-)] and added to the wells. After incubation at 4 °C overnight, the wells were washed and incubated for 2 h at room temperature with 50 μL of horse radish peroxidase (HRP)-conjugated sheep anti-mouse IgG (1:8000, GE Healthcare, NA931), HRP-conjugated goat anti-mouse IgG1 (1:10,000, Bethyl, A90-105P), HRP-conjugated goat anti-mouse IgG2a (1:10,000, Bethyl, A90-107P), or HRP-conjugated goat anti-monkey IgG (1:8000, NORDIC IMMUNOLOGY, GAMon/IgG(Fc)/PO) antibodies. After washing the wells, 100 μL of a mixture of *o*-phenylenediamine dihydrochloride and hydrogen peroxide (H_2_O_2_) in citrate–phosphate buffer was added to each well. The reactions were quenched by adding 1 M sulfuric acid to each well, and the absorbance was measured at 492 nm. To calculate the ratio of IgG2a/IgG1 specific to H5 HA in the sera of mice vaccinated with rDIs-mcl2.2 HA, endpoint titers of IgG2a and IgG1 specific to H5 HA were determined. The endpoint titer was defined as the reciprocal of the highest dilution of serum at which the absorbance at 490 nm exceeded two-fold the value of the blank.

### 2.8. In Vitro Neutralization Titer Assay for VACV

The sera of mice were inactivated using heat at 56 °C for 30 min. Serial dilutions of heat-inactivated sera were mixed with equal volumes of 1000 PFU/mL of LC16m8, and then incubated at 37 °C for 1 h, followed by incubation at 4 °C for 16 h. RK13 cells were then infected with 100 μL of the virus/serum mixtures in 6-well plates. At 72 h after infection, the neutralization titer was expressed as the reciprocal of the maximum dilution of serum at which the plaque numbers were reduced by 50% compared with that in wells infected with the virus in the absence of serum.

### 2.9. Hemagglutination Inhibition (HI) Assay

HI assays were performed according to standard methods [23]. Briefly, serum samples were pretreated (37 °C, 18 h) with a receptor-destroying enzyme (RDE II, Denka Seiken, Tokyo, Japan) and then inactivated using heat (56 °C, 30 min). The resulting serum samples were subjected to serial two-fold dilutions with PBS(-) and then mixed with a 4 × hemagglutination (HA) titer of H5N1 A/whooper swan/Hokkaido/1/2008, A/whooper swan/Mongolia/3/2005, or A/Vietnam/UT3040/2004 in a 0.75% suspension of guinea pig erythrocytes. After 1 h of incubation at room temperature, hemagglutination was determined by visual inspection. Titers were expressed as the reciprocal of the maximum dilution of serum that completely inhibited hemagglutination.

### 2.10. ELISpot Assay

Four weeks after a single intradermal inoculation of rDIs-mcl2.2 HA into eight-week-old female BALB/c mice, splenocytes were harvested and stored in liquid nitrogen. Thawed splenocytes [2 × 10^5^ cells/well] were seeded into the wells of 96-well MultiScreen IP Sterile plates (Merck Millipore Ltd., Burlington, MA, USA) coated with anti-IFN-γ (MABTECH, Nacka strand, Sweden #3321-2H) or -IL-4 (MABTECH, Nacka strand, Sweden #3311-2H) antibody and subsequently stimulated with pooled H5 HA peptide (PepMix Influenza A (HA/Indonesia [H5N1]) [PM-INFA-HAIndo, JPT Peptide Technologies]) at 37 °C for 48 h. The production of IFN-γ and IL-4 was analyzed using an ELISpot assay kit (MABTECH, Nacka Strand, Sweden) according to the manufacturer’s instructions. After drying the ELISpot plates, the number of spots in each well was counted using an automated ELISpot plate reader (Advanced Imaging Devices GmbH, Strassberg, Germany).

### 2.11. Lung Histopathology and Inflammation Scores

Small pieces of all lung lobes of cynomolgus macaques were fixed in 10% neutral-buffered formalin, embedded in paraffin, sectioned at a thickness of 4 μm, stained with hematoxylin and eosin (H-E), and then subjected to routine histological examination. Histopathological findings were evaluated according to the modified methods described in a previous study [17]. The grading system for the histological changes was as follows: For each lung tissue section, 10 randomly selected microscopic fields were scanned at a magnification of 100×, and each field was graded visually on a scale from 0 to 7: 0, normal lung; 1, mild destruction of epithelium in trachea and bronchus; 2, mild infiltration of inflammatory cells around the periphery of bronchiole; 3, moderate infiltration of inflammatory cells around the alveolar walls, resulting in alveolar thickening; 4, mild alveolar injury accompanied by vascular damage (<10%); 5, moderate alveolar and vascular injury (10–30%); 6, severe alveolar injury with hyaline membrane-accompanied alveolar hemorrhage (<50%); and 7, severe alveolar injury with hyaline membrane accompanied alveolar hemorrhage (>50%). The mean value of the grades obtained for all fields from a given animal was used as the grade of visual lung injury.

### 2.12. Viral RNA Quantification

All lung lobes of the cynomolgus macaques were homogenized in nine volumes of Leibovitz’s L-15 medium (Thermo Fisher Scientific, Waltham, MA, USA) using a BioMasher II (Nippi, Tokyo, Japan). Total RNA was extracted from the homogenates using QIAamp Viral RNA Mini kits (Qiagen, Hilden, Germany) according to the manufacturer’s instructions. Viral RNA was quantified by quantitative reverse transcription PCR (qRT-PCR), as previously described [24].

### 2.13. Statistical Analyses

Data are expressed as mean ± standard deviation (SD) or geometric mean ± geometric SD. Inferential statistical analysis was performed using a two-tailed non-paired Student’s *t*-test, Mann–Whitney U test, or one-way ANOVA followed by Tukey’s test, as appropriate. The Gehan–Breslow–Wilcoxon method was used for statistical analysis of survival rates. *p* values < 0.05 were considered statistically significant. For all statistical analyses, the Prism software package was used (version 8.4 and 10.4.0; GraphPad Software).

## 3. Results

### 3.1. Single Dose of Recombinant VACV LC16m8 Encoding H5 Hemagglutinin Protein Protects Mice from Lethal Challenges with the HPAI H5N1 Virus

We generated two recombinant H5N1 vaccine candidates based on the attenuated VACV LC16m8 strain by inserting a gene encoding an H5 HA protein without multi-basic sites. We employed genes encoding H5 HA proteins derived from either clade 2.2 or clade 2.3.4, yielding vaccine candidates designated as rLC16m8-mcl2.2 HA and -mcl2.3.4 HA, respectively (Appendix A). To investigate the protective efficacy of these two H5N1 vaccine candidates, BALB/c mice were intradermally immunized with 1 × 10^7^ PFU of either rLC16m8-mcl2.2 HA or -mcl2.3.4 HA (Figure 1A). Control mice were inoculated with recombinant LC16m8-empty, which consisted of the vector including the ATI/p7.5 hybrid promoter sequence, without any inserted influenza virus genes. Five weeks after vaccination, the mice were intranasally infected with a lethal dose [1 × 10^4^ PFU; 166× 50% mouse lethal dose (MLD_50_); Appendix A] of the HPAI A/whooper swan/Hokkaido/1/2008 (H5N1) virus (clade 2.3.2.1). Closely related clade vaccine rLC16m8-mcl2.3.4 HA-immunized mice recovered rapidly from a loss of body weight following HPAI H5N1 virus infection, and all mice survived (Figure 1B,C). The pulmonary H5N1 viral titers in the rLC16m8-mcl2.3.4 HA-immunized mice were significantly lower than those in the control mice (rLC16m8-empty) at 1 dpi and decreased to close to or below the detection limit by 6 dpi (Figure 1D). Importantly, the heterologous clade vaccine rLC16m8-mcl2.2 HA-immunized mice also recovered rapidly from the loss of body weight and showed a 100% survival rate (Figure 1B,C). The pulmonary viral titers in rLC16m8-mcl2.2 HA-immunized mice decreased below the detection limit by 6 dpi. These results indicate that the protective efficacy of the heterologous clade vaccine rLC16m8-mcl2.2 HA against clade 2.3.2.1 HPAI H5N1 virus is comparable to or greater than that of the closely related clade vaccine rLC16m8-mcl2.3.4 HA.

We investigated whether rLC16m8-mcl2.2 HA could protect mice that had been sensitized previously with VACVs from lethal challenges with the HPAI H5N1 virus. BALB/c mice were inoculated with 1 × 10^7^ PFUs of VACV (rLC16m8-empty). Subsequently, 4 weeks later, the mice were inoculated with 1 × 10^7^ PFUs of rLC16m8-mcl2.2 HA. Four weeks after rLC16m8-mcl2.2 HA inoculation, a lethal challenge with the HPAI H5N1 virus was conducted. In terms of both the change in body weight and survival rate, no significant difference was noted in the protective efficacy of rLC16m8-mcl2.2 HA between mice pre-sensitized with VACV and naïve mice inoculated with a vehicle instead of VACV (Figure 1E,F). Taken together, these results demonstrate that rLC16m8-mcl2.2 HA showed cross-clade protection against lethal infection with the heterologous clade HPAI H5N1 virus, even in mice pre-sensitized with VACV.

### 3.2. Rapid Protection in Mice by rLC16m8-H5 HA Vaccine Candidates

Next, we investigated the rapidity of protection conferred by rLC16m8-H5 HA against the HPAI H5N1 virus. BALB/c mice were inoculated intradermally with 1 × 10^7^ PFUs of either rLC16m8-mcl2.2 HA or –mcl2.3.4 HA; these inoculations were performed 1 or 2 weeks or 3 days prior to a lethal challenge with A/whooper swan/Hokkaido/1/2008 (Figure 2A). All vaccinated mice recovered rapidly from the loss of body weight and survived a lethal challenge with the HPAI H5N1 virus when the vaccine was administered 1 or 2 weeks prior to the H5N1 virus challenge (Figure 2B,C). In contrast, no protective effect was observed when the rLC16m8 H5 HA vaccine was administered 3 days prior to HPAI H5N1 virus infection (Figure 2D). Given that rLC16m8-H5 HA vaccines require at least 1 week to elicit protective immunity, we infer that adaptive immune responses play a crucial role in the protective effects exerted by these vaccines against the HPAI H5N1 virus.

### 3.3. Long-Term Protection in Mice by rLC16m8-H5 HA Vaccine Candidates

To investigate the duration over which the immunity induced by the rLC16m8-mcl2.2 HA vaccine was maintained, BALB/c mice were inoculated intradermally with rLC16m8-mcl2.2 HA, followed by antibody responses against the H5 HA antigen at subsequent time points for 20 months. After vaccination, the induction of IgG specific to H5 HA (clade 2.2) was monitored using an ELISA. The serum level of clade 2.2 HA antigen-specific IgG increased before reaching a plateau at seven months. The resulting high serum IgG levels were retained during the 20-month experimental period (Figure 3A). In contrast, no HA-specific IgG was detected in the control mice immunized with rLC16m8-empty. HI activities against three different clades of the H5N1 virus in the sera of rLC16m8-mcl2.2 HA-immunized mice were also higher 6 months after vaccination than those recorded 1 month after vaccination (Figure 3B). Next, we examined the long-term protective efficacy of the rLC16m8-mcl2.2 HA vaccine. A total of 12 or 18 months after vaccination, the mice were intranasally infected with a lethal dose of H5N1 A/whooper swan/Hokkaido/1/2008. Control mice showed a rapid loss of body weight before succumbing to the infection (Figure 3C,D). In contrast, all vaccinated mice recovered rapidly from body weight loss and survived up to 18 months post-vaccination (Figure 3C,D). The protective efficacy of rLC16m8-mcl2.2 HA at 18 months post-vaccination was comparable to that of rLC16m8-mcl2.2 HA at 5 weeks post-vaccination (Figure 1B,C). Furthermore, rLC16m8-mcl2.2 HA protected mice from lethal challenge with another H5N1 clade virus (A/Vietnam/UT3040/2004), even when infected 20 months after vaccination (Figure 3E). These results suggest that rLC16m8-mcl2.2 HA can elicit a broad cross-clade immunity against the HPAI H5N1 virus throughout its lifetime.

### 3.4. Long-Term Protection in Non-Human Primates by rLC16m8-mcl2.2 HA Vaccine Candidates

Next, we examined the long-term protection in cynomolgus macaques vaccinated with rLC16m8-mcl2.2 HA. Female cynomolgus macaques were inoculated with a per-animal dose of 1 × 10^7^ PFUs of rLC16m8-mcl2.2 HA or rLC16m8–empty, administered intradermally in both upper arms. The production of antibodies specific to H5 HA (clade 2.2) was monitored over the subsequent 12 months via an ELISA. In rLC16m8-mcl2.2 HA-immunized macaques, the levels of clade 2.2 HA Ag-specific IgG gradually increased over the experimental period (12 months post-vaccination; Figure 4A). A total of 12 months after immunization with a single dose of rLC16m8-mcl2.2 HA, the macaques were infected with the HPAI H5N1 virus (A/whooper swan/Hokkaido/1/2008). Body temperature changes were measured in infected macaques every night (8 p.m. to 8 a.m.) and compared with those obtained before viral infection (day 0). All infected macaques had a fever on the night of the H5N1 virus infection, and high temperatures persisted for the following two nights in three of the five infected macaques (Figure 4B). Thereafter, unvaccinated macaques (#35 and #37) maintained a high fever for the remaining experimental interval (until 7 dpi). In contrast, the body temperatures of the vaccinated macaques (#38, #39, and #40) decreased after 3 dpi and recovered to normal levels, indicating that fever, especially in the late phase, was suppressed in the vaccinated macaques (Figure 4C). We measured temporal changes in viral titers in swab samples collected from the conjunctiva, nasal cavity, oral cavity, trachea, and rectum. Infectious HPAI H5N1 viruses were detected in the nasal, oral, and tracheal cavities (Figure 4D and Appendix A). Notably, while infectious viruses were detected in unvaccinated macaques until 5 dpi (Figure 4D), macaques in the vaccinated group eliminated the infectious viruses earlier (Figure 4D).

Pneumonia progression was assessed using chest radiography (Appendix A). Unvaccinated macaques exhibited abnormal densities in both lungs at 7 dpi. Of the rLC16m8-mcl2.2 HA-immunized animals, #38 showed a slightly abnormal density in the left lung, #39 did not show a significant change in lung density, and #40 showed an abnormal density in both the right and left lungs. Infected macaques were euthanized on day seven after infection, and histopathological analysis was performed (Figure 5A). Consistent with the radiographic analyses, unvaccinated macaques showed severe inflammation in several lobes of both lungs. In contrast, two out of three vaccinated macaques showed alleviation of pneumonia (Figure 5B). The viral load in the lungs was significantly lower in the vaccinated group than in the unvaccinated group (Figure 5C), indicating that all vaccinated macaques eliminated the H5N1 virus earlier. These results demonstrate that a single dose of the recombinant H5 HA vaccine based on attenuated VACV LC16m8 provided cross-clade protection against heterologous HPAI H5N1 viruses in mice and macaques.

### 3.5. Rapid Protection and Long-Term Protection Against HPAI H5N1 Virus Induced by a Single Dose of rDIs-mcl2.2 HA in Mice

We also generated a recombinant H5 HA vaccine candidate based on the VACV DIs strain, a mutant with a highly restricted host range [19]. To evaluate the potential of the DIs-based recombinant vaccine as a vaccine platform, we investigated the protective efficacy of rDIs-mcl2.2 HA against HPAI H5N1 viral infection in a mouse model. A single dose of rDIs-mcl2.2 HA protected mice from lethal challenge with a heterologous clade HPAI H5N1 virus when the vaccine was used for immunization as little as 1 week prior to the challenge infection, ensuring the survival of all vaccinated mice (Figure 6A–C). In contrast, mice in the unvaccinated group, which were mock-inoculated or inoculated with DIs, showed a rapid loss of body weight and succumbed to the lethal HPAI H5N1 virus infection. To investigate long-term immunity following rDIs-mcl2.2 HA vaccination, we measured H5 HA-specific IgG production by an ELISA (Figure 6D). The serum levels of H5 HA-specific IgG increased gradually and peaked seven months after vaccination. Importantly, the high serum levels of IgG were retained even 20 months after vaccination, when the IgG level was comparable to that observed in rLC16m8-mcl2.2 HA-vaccinated mice (Figure 3A). Furthermore, HI activities against three different clades of the H5N1 virus in the sera of rDIs-mcl2.2 HA-immunized mice were higher 6 months after vaccination than those recorded 1 month after vaccination (Figure 6E), similar to those observed with rLC16m8-mcl2.2 HA (Figure 3B). When the rDIs-mcl2.2 HA-vaccinated mice were challenged with a lethal dose of HPAI H5N1 virus 20 months after vaccination, none of the infected mice showed body weight loss, and all animals survived the infection (Figure 6F). In contrast, unvaccinated mice showed a rapid loss of body weight and succumbed to infection within eight days after infection (Figure 6F). Thus, the rDIs-mcl2.2 HA vaccine afforded rapid and long-term protection against HPAI H5N1 viral infection, similar to rLC16m8-mcl2.2 HA. These results suggest that replication-deficient DIs can serve as platforms for the construction of recombinant live vaccines for use in humans.

### 3.6. Protective Efficacy of rDIs-mcl2.2 HA Against HPAI H5N1 Virus in VACV-Sensitized Mice

Next, we investigated the protective efficacy of a single dose of rDIs-mcl2.2 HA against lethal infection with the HPAI H5N1 virus in VACV-sensitized mice. BALB/c mice were divided into two groups: a naïve group and a VACV-sensitized group, inoculated intradermally with 1 × 10^7^ PFU of VACV LC16m8 before vaccination with rDIs-mcl2.2 HA (Figure 7A). Twenty-one weeks after VACV sensitization, the mean neutralization titer (50% neutralization) against LC16m8 was higher than 1:30 (corresponding to dilution with geometric mean and geometric SD of 31.32 ± 7.61; Figure 7B), indicating that the neutralization titer was comparable to that in human populations [25,26]. In the VACV-sensitized group, mice were divided into four subgroups, in which the neutralization titer against VACV was similar within each subgroup (Appendix A). Both naïve mice and VACV-sensitized mice were immunized intradermally with 1 × 10^7^, 3 × 10^7^, or 1 × 10^8^ PFU of rDIs-mcl2.2 HA 22 weeks after VACV pre-immunization, and all mice were infected with a lethal dose of HPAI H5N1 virus 5 weeks after vaccination. When naïve mice were vaccinated, 1 × 10^7^ PFUs of rDIs-mcl2.2 HA were sufficient to completely protect the mice from lethal infection with the H5N1 virus (Figure 7C). In mice pre-sensitized with VACV, the protective efficacy of rDIs-mcl2.2 HA against HPAI H5N1 virus infection increased with an increasing dose of vaccine (Figure 7D). When the mice were immunized with 1 × 10^8^ PFUs of rDIs-mcl2.2 HA, 89% (8 of 9) of the infected mice survived the challenge (Figure 7D, right panel).

### 3.7. Th1/Th2 Immune Response to rVACV-mcl2.2 HA Vaccine

Finally, we investigated the Th1/Th2 balance against rVACV-mcl2.2 HA because of the risk of worsening symptoms during subsequent pathogen infection associated with vaccine-induced immune responses, especially those with a Th1/Th2 balance skewed toward Th2. Antisera from mice immunized with rLC16m8-mcl2.2 HA or rDIs-mcl2.2 HA (1 and 6 months post-vaccination) were used to measure the endpoint titers of IgG1 and IgG2a specific to H5 HA using an ELISA. The rLC16m8-mcl2.2 HA and rDIs-mcl2.2 HA vaccines potently induced both IgG1 and IgG2a (Figure 8A), and the IgG2a/IgG1 ratio indicated a Th1-dominant phenotype after 1 and 6 months (Figure 8B). A Th1-dominant response was also confirmed by antigen-specific T-cell responses in ELISpot assays. Mice vaccinated with rDIs-mcl2.2 HA showed significant production of IFN-γ upon antigen stimulation, whereas the IL-4 level was comparable to that in controls (Figure 8C). Taken together, these results suggest that both the rLC16m8-mcl2.2 HA and rDIs-mcl2.2 HA vaccines induce a Th1-dominant or balanced Th1/Th2 response.

## 4. Discussion

In this study, we generated recombinant live vaccines encoding an H5-subtype HA protein using two different strains of highly attenuated VACV: LC16m8 and DIs. We found that a single dose of recombinant LC16m8 expressing H5 HA from clade 2.2 (rLC16m8-mcl2.2 HA) or clade 2.3.4 (rLC16m8-mcl2.3.4 HA) protected mice from lethal challenge with the HPAI H5N1 virus from clade 2.3.2.1, and mice vaccinated with rLC16m8-mcl2.2 HA eliminated the pulmonary infectious virus more rapidly than mice vaccinated with rLC16m8-mcl2.3.4 HA (Figure 1D). In subsequent experiments, we focused primarily on the protective efficacy of clade 2.2 HA-expressing recombinant VACV vaccines. Notably, a single dose of either rLC16m8-mcl2.2 HA or rDIs-mcl2.2 HA in mice provided rapid (as early as 1 week after vaccination) and long-term (as long as 20 months after vaccination) protection against heterologous HPAI H5N1 viruses (clade 1 or 2.3.2.1). Genetic diversification of H5 viruses into various clades and subclades has increased the antigenic diversity of HA [27,28,29]. It is difficult to predict which subtypes and clades of the influenza virus will cause future pandemics. Therefore, it is crucial to develop vaccines that exhibit broad cross-reactivity against various strains of a given serotype, such as the H5 subtype.

Unlike seasonal influenza vaccines, an influenza vaccine designed specifically for a pandemic is used when an influenza outbreak is imminent; hence, such a vaccine would need to elicit a rapid protective immune response in unexposed individuals. The H1N1 pandemic influenza virus, which emerged in April 2009, spread globally, resulting in more than 70,000 laboratory-confirmed cases and 300 deaths in over 100 countries within 6 weeks [30]. Given the global rapid spread of these novel influenza viruses, pre-pandemic vaccines must rapidly induce a strong immune response specific to the pandemic virus. An inactivated H5N1 whole-virion vaccine has been manufactured and stored in many countries as a pre-pandemic vaccine, and its safety profile renders it particularly appealing for use [31,32]. However, such pre-pandemic vaccines require multiple immunizations to induce specific immune responses against HPAI H5N1 viruses; thus, such a vaccine will take longer than four weeks to induce an optimal immune response against the HPAI H5N1 pandemic virus. In addition, there are concerns that the protective efficacy of these inactivated H5N1 vaccines is significantly lower against pandemic strains whose antigenicity does not match that of the vaccine virus strains [33]. Recently, a vaccine platform to develop cell culture-derived vaccines that antigenically match a pandemic virus was established, rendering the provision of an antigenically matched vaccine globally within 6 months of initiating such a project possible [34]. However, there remains an inevitable interval between the emergence of a novel influenza virus and the supply of an antigen-matched vaccine. Under such circumstances, both rLC16m8- and rDIs-H5 HA vaccines that provide rapid protection (as early as 1 week after vaccination) against lethal challenges with heterologous HPAI H5N1 viruses (Figure 2C and Figure 6C) would be useful.

During pandemics, there is an urgent need for vaccination against the responsible strain. However, influenza viruses circulate perpetually in aquatic birds, and certain subtypes of influenza viruses can reappear at low frequencies (after several decades of absence) in humans. In addition, when outbreaks of new subtypes of influenza viruses, such as the pandemic H1N1 virus and avian H7N9 virus, are noted in humans, these viruses may continue to circulate. Considering that human cases of HPAI H5N1 virus infection have been reported over the 15 years since the first reported human infection by the virus in Hong Kong in 1997 [35], and given that H5-subtype HPAI viruses, including the H5N1 virus, are still circulating in wild aquatic birds and domestic poultry worldwide, the induction of long-lasting immunity is also important for controlling human cases of infection with the HPAI H5N1 virus. One case of human infection with the HPAI H5N1 virus was reported in Nepal in 2019 [36]. In general, currently stockpiled vaccines do not confer long-term immunity against H5N1 viruses, even when administered in the presence of an adjuvant. In contrast, we observed that rLC16m8- and rDIs-mcl2.2 HAs conferred long-lasting immunity in the animal models (Figure 3, Figure 4, Figure 5 and Figure 6). These results are consistent with the fact that individuals immunized with VACV retain substantial antigen-specific immunity for more than 25 years after the end of the vaccination [25,37]. In addition, regarding antibody production, the immune response was stronger approximately 6 months after rVACV vaccination than that recorded after one month (Figure 3A,B, Figure 4A and Figure 6D,E). Thus, even with a recurrent outbreak of the HPAI H5N1 virus in the future, previous immunization with rLC16m8- or rDIs-mcl2.2 HA might contribute to preventing the spread of such an emerging viral strain in subsequent decades.

In this study, we demonstrated that a single vaccination with rVACV-H5 HA could confer rapid and long-term protection against H5N1 viral infections in animals. We also demonstrated that vaccination with rVACV-H5 HA can induce both antigen-specific antibody production and T-cell responses that are Th1-dominant or Th1/Th2-balanced. However, the involvement of induced antibodies and cellular immunity in the protective effect of rVACV-H5 HA vaccination requires further study, such as passive transfer of antigen-specific antibodies, adoptive transfer of T cells, or T-cell depletion experiments.

Importantly, the protective efficacy of rDIs-mcl2.2 HA was comparable to that of rLC16m8-mcl2.2 HA, even though the DIs strain had a restricted host range and could not produce progeny virions in most mammalian cells [19]. Currently, only a few highly attenuated VACVs exhibit restricted host ranges. These host-range-restricted viruses include DIs and modified vaccinia Ankara (MVA), which have been generated from a parental strain (CVA) by multiple passages in CEFs [38]. MVA has also been intensively used as a platform for the development of vaccines against various infectious diseases, including the HPAI H5N1 virus. One or two doses of rMVA-H5 HA demonstrate complete protection against lethal challenges with homologous and heterologous H5N1 HPAI viruses in mice, inducing B-cell immunity [39]. rMVA-H5 HA vaccines have been investigated in clinical trials [40,41]. However, rapid and long-term protection by rMVA-H5 HA remains to be elucidated. It would be valuable to determine whether the protective potential against H5N1 HPAI viruses differs between DIs- and MVA-based vaccines, given that the sites and sizes of deletions differ between DIs and MVA. Here, we demonstrated that rDIs-mcl2.2 HA afforded lifelong protection against the H5N1 HPAI virus in mice. Although lifelong immunity has been hypothesized to require infection by replication-competent viruses, our data suggest that a non-replicating viral vector, such as DIs, might also provide lifelong immunity in individuals. Given the safety of such derivatives, non-replicating viruses should be investigated for use as viral vectors in the development of recombinant vaccines.

As mentioned previously, many VACV strains, including Lister, Copenhagen, CVA, and New York City Board of Health (NYCBH), were used during the WHO smallpox eradication campaign [42]. Therefore, worldwide, most people over the age of 50 are immune to VACV. Inducing immunity against an antigen of interest using a recombinant vaccine based on a VACV vector in such individuals may pose a significant challenge. To address this issue, we investigated the protective efficacy of rLC16m8-mcl2.2 HA and rDIs-mcl2.2 HA against HPAI H5N1 viral challenge in VACV-sensitized mice. Protection of VACV-sensitized animals from lethal challenge with the HPAI H5N1 virus was achieved not only with rLC16m8-mcl.2.2 HA but also with rDIs-mcl2.2 HA; however, a higher inoculation titer of rDIs-mcl2.2 HA than rLC16m8-mcl2.2 HA was required (Figure 1F and Figure 7D). These results indicate that both H5 vaccine candidates may be efficacious for all generations, including populations with preexisting immunity to VACV.

## 5. Conclusions

A single dose of either of the two H5 HA-expressing VACV vaccines based on rLC16m8 or rDIs vectors provided both rapid and long-lasting protection against heterologous H5N1 HPAI viruses in mice and cynomolgus macaques. These recombinant H5-subtype vaccines based on highly attenuated VACV vectors are promising candidates for future use against H5-subtype influenza viruses, including the H5N1 HPAI virus. In particular, a DIs-based vaccine that employs a replication-deficient vector may be safer for use in populations that include immunocompromised individuals.

## Figures and Tables

**Figure 1 vaccines-13-00074-f001:**
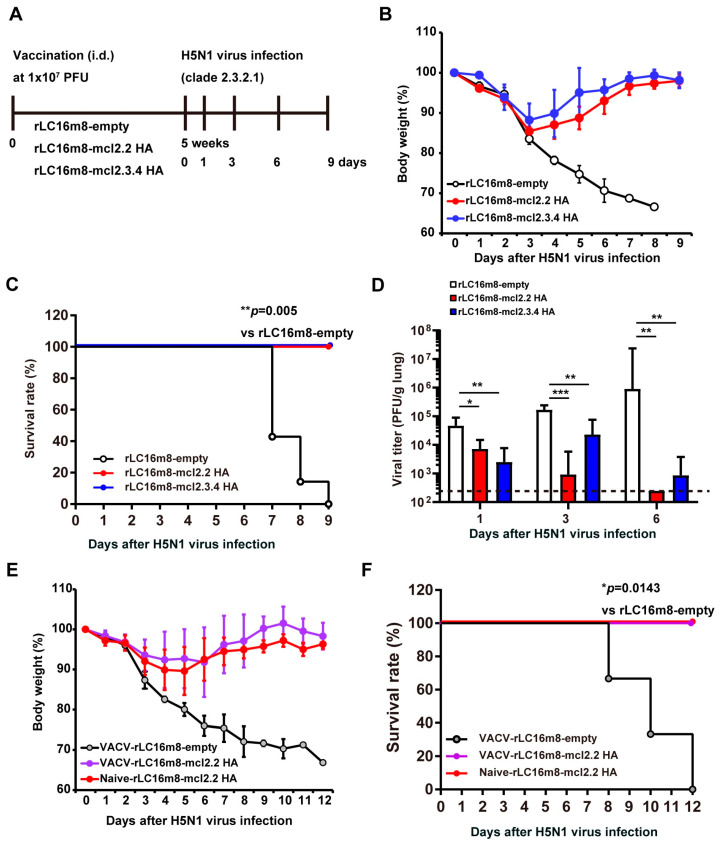
Protective efficacy of recombinant vaccinia virus LC16m8 strain (rLC16m8) expressing H5 hemagglutinin (HA) protein against lethal infection with highly pathogenic avian influenza (HPAI) H5N1 virus in mice. (**A**–**D**) Vaccination and infection studies in naïve BALB/c mice. (**A**) Experimental schedule of vaccination and H5N1 influenza virus infection in mice. Eight-week-old female BALB/c mice were inoculated intradermally with 1 × 10^7^ PFU of rLC16m8-mcl2.2 HA, rLC16m8-mcl2.3.4 HA, or rLC16m8-empty; 5 weeks after vaccination, animals were infected intranasally with 1 × 10^4^ PFU of H5N1 A/whooper swan/Hokkaido/1/2008 (clade 2.3.2.1). rLC16m8-mcl2.2 HA, rLC16m8 encoding the H5 HA protein (clade 2.2, A/Qinghai/1A/05); rLC16m8-mcl2.3.4 HA, rLC16m8 encoding the H5 HA protein (clade 2.3.4, A/Anhui/1/05); rCL16m8-empty, rCL16m8 harboring only the ATI/p7.5 hybrid promoter sequence. (**B**) Body weight was monitored daily after the H5N1 virus infection. Values are shown as mean ± SD. (**C**) Survival rate was observed until 9 days post-infection (dpi). (**D**) Pulmonary virus titer was determined in four mice per group at each time point after H5N1 influenza virus infection. Values are shown as geometric mean ± geometric SD. *p* values are calculated using two-tailed non-paired one-way ANOVA followed by Turkey’s test. (**E**,**F**) Vaccination and infection studies in BALB/c mice sensitized with vaccinia virus (VACV). Eight-week-old female BALB/c mice were inoculated intradermally with either 1×10^7^ PFU of rLC16m8-empty or culture medium (vehicle). Four weeks after sensitization with VACV, these mice were further inoculated with 1 × 10^7^ PFU of either rLC16m8-mcl2.2 HA or rLC16m8-empty. Four weeks after vaccination with rLC16m8-based viruses, animals were infected with 1 × 10^4^ PFU of H5N1 A/whooper swan/Hokkaido/1/2008. * *p* < 0.05, ** *p* < 0.01, *** *p* < 0.001. (**E**) Body weight was monitored daily after H5N1 virus infection. Values are shown as mean ± SD. (**F**) Survival rate was observed until twelve dpi. Survival rates were compared in panels (**C**,**F**) using the Gehan-Breslow-Wilcoxon method.

**Figure 2 vaccines-13-00074-f002:**
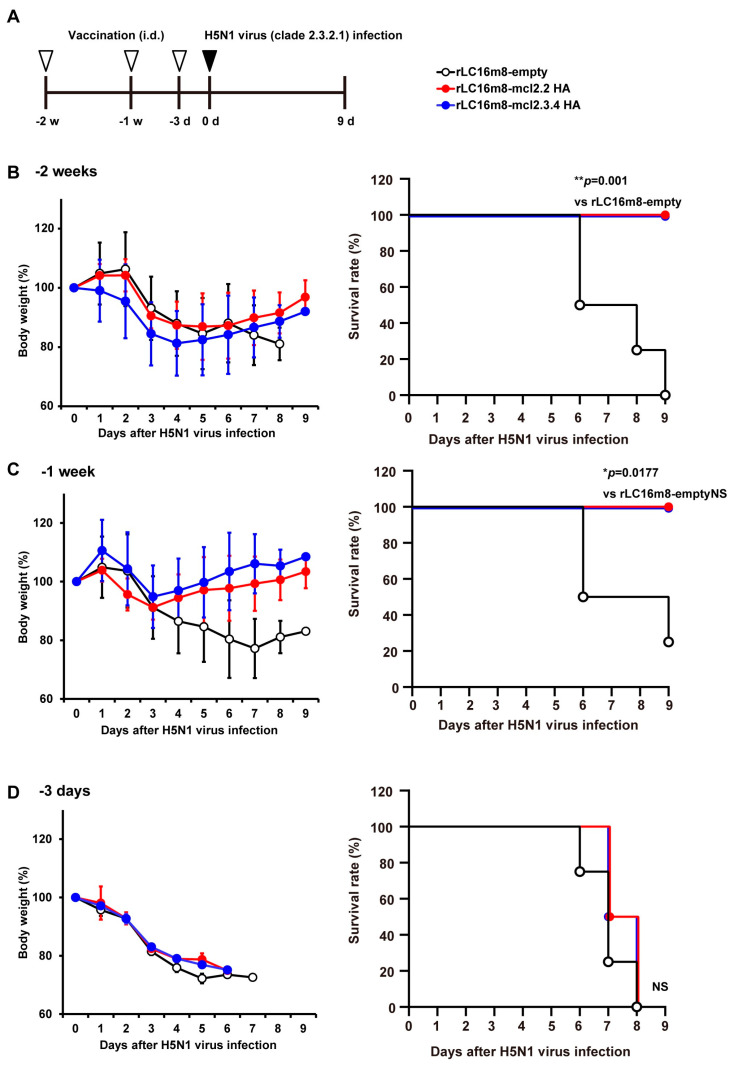
Rapid protective effects induced by rLC16m8-H5 HA against lethal infection with the HPAI H5N1 virus in mice. (**A**) Experimental schedule of vaccination and H5N1 influenza virus infection in BALB/c mice. Female BALB/c mice were inoculated intradermally with 1 × 10^7^ PFUs of rLC16m8-mcl2.2 HA, rLC16m8-mcl2.3.4 HA, or rLC16m8-empty (**B**) 2 weeks, (**C**) 1 week, or (**D**) 3 days before intranasal infection with 1 × 10^4^ PFUs of H5N1 A/whooper swan/Hokkaido/1/2008 (clade 2.3.2.1). (**B**) BALB/c mice were infected with the H5N1 HPAI virus 2 weeks after vaccination with rLC16m8-H5 HA. After infection with the virus, body weight was monitored daily (left panel), and the survival rate was recorded until 9 days post-infection (dpi) (right panel). (**C**) One week after vaccination, the rapid protective efficacy of rLC16m8-H5 HA against H5N1 HPAI virus infection was evaluated by monitoring body weight daily (left panel) and recording survival until 9 dpi (right panel). (**D**) Three days after vaccination, the rapid protective efficacy of rLC16m8-H5 HA against H5N1 HPAI virus infection was evaluated by monitoring body weight daily (left panel) and recording survival until 9 dpi (right panel). Survival rates are compared in panels (**B**–**D**) using the Gehan–Breslow–Wilcoxon method. NS: not significant. * *p* < 0.05, ** *p* < 0.01.

**Figure 3 vaccines-13-00074-f003:**
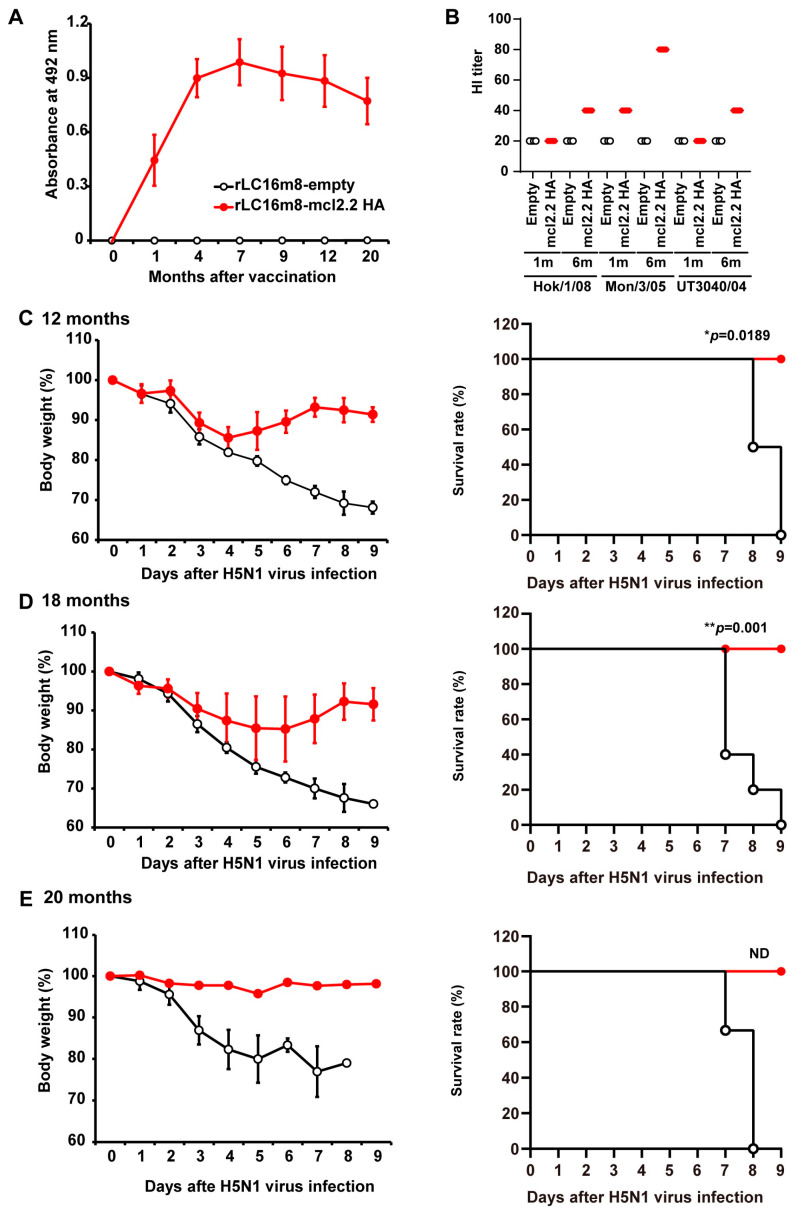
Long-term protection by rLC16m8-H5 HA against lethal infection with the HPAI H5N1 virus in mice. Female BALB/c mice were inoculated intradermally with 1 × 10^7^ PFUs of rLC16m8-mcl2.2 HA or rLC16m8-empty [*n* = 3–5 in each group except for *n* = 2 in rLC16m8-mcl2.2 HA in (**E**)]. (**A**) The time course of production of immunoglobulin G (IgG) specific to H5 HA (clade 2.2) after vaccination was evaluated by an ELISA. (**B**) The HI titer of antisera from mice immunized with rLC16m8-mlc.2.2 HA (*n* = 4) or rLC16m8-empty (*n* = 4) 1 and 6 months after immunization. HI titers were determined against HPAI H5N1 A/whooper swan/Hokkaido/1/2008 (clade 2.3.2.1; Hok/1/08), A/whooper swan/Mongolia/3/2005 (clade 2.2; Mon/3/05), and A/Vietnam/UT3040/2004 (clade 1; UT3040/04) using 0.75% guinea pig erythrocytes. (**C**–**E**) Vaccinated mice were challenged with 166 × 50% mouse lethal dose (MLD_50_) of HPAI H5N1 A/whooper swan/Hokkaido/1/2008 (clade 2.3.2.1) 12 (**C**) or 18 (**D**) months after vaccination. (**E**) Vaccinated mice were challenged with 150 × MLD_50_ of H5N1 A/Vietnam/UT3040/2004 (clade 1) 20 months after vaccination. Left panels show body weight changes when monitored daily after the H5N1 virus challenge. Right panels show the survival rate when assessed until 9 days post-infection. ND: not determined (rLC16m8-mcl2.2 HA, *n* = 2; rLC16m8-empty, *n* = 3). * *p* < 0.05, ** *p* < 0.01.

**Figure 4 vaccines-13-00074-f004:**
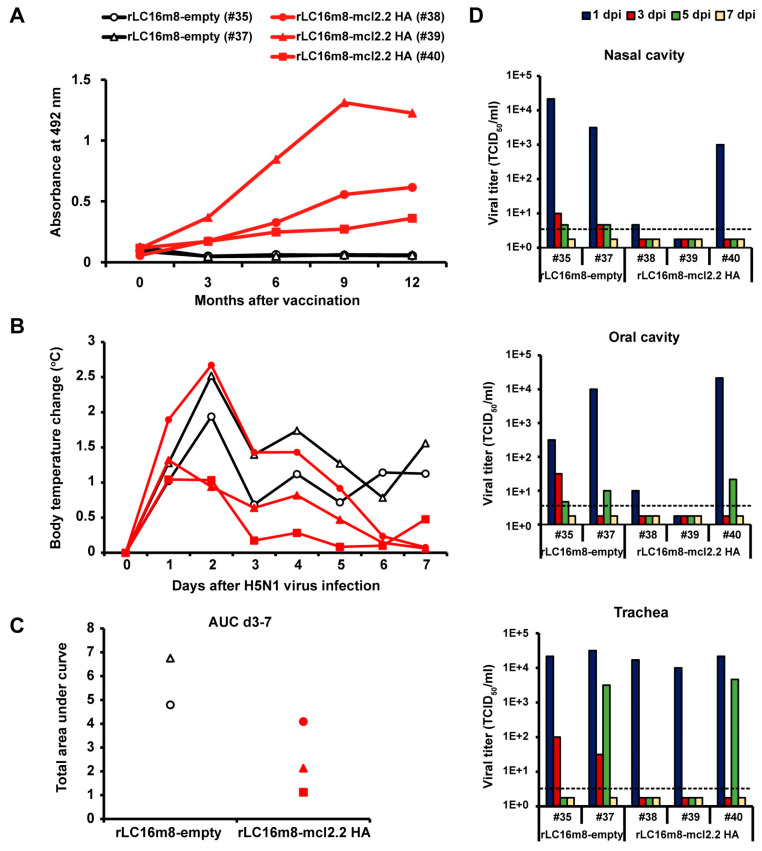
Long-term protection by rLC16m8-H5 HA against lethal infection with the HPAI H5N1 virus in cynomolgus macaques. Female cynomolgus macaques were inoculated intradermally with 1 × 10^7^ PFUs of rLC16m8-mcl2.2 HA (*n* = 3) or rLC16m8-empty (*n* = 2) on their upper arms. Twelve months after vaccination, HPAI H5N1 virus A/whooper swan/Hokkaido/1/2008 was inoculated into the nostrils, oral cavity, and trachea of each macaque. (**A**) The time course of production of IgG specific to the H5 HA protein (clade 2.2) after vaccination was evaluated using an ELISA. (**B**) The mean value of the body temperature of individual macaques from 8 p.m. to 8 a.m. every night was calculated from the temperature recorded every 5 min. Body temperature changes of individual macaques on each day after virus infection were compared with mean temperature changes from 8 p.m. on day 1 to 8 a.m. on day 0 before virus infection. (**C**) Cumulative temperature increase, calculated as the area under the curve (AUC) from the data recorded 3 days post-infection (dpi) to 7 dpi in (**B**). (**D**) Temporal changes in viral titers in nasal (upper panel), oral (middle panel), and tracheal (lower panel) swab samples were determined by a 50% tissue culture infectious dose (TCID_50_) assay using Madin–Darby canine kidney (MDCK) cells. The dpis are indicated in navy (1 dpi), red (3 dpi), green (5 dpi), and yellow (7 dpi). The lower limit of detection (1.7 log units) is indicated by a horizontal dashed line.

**Figure 5 vaccines-13-00074-f005:**
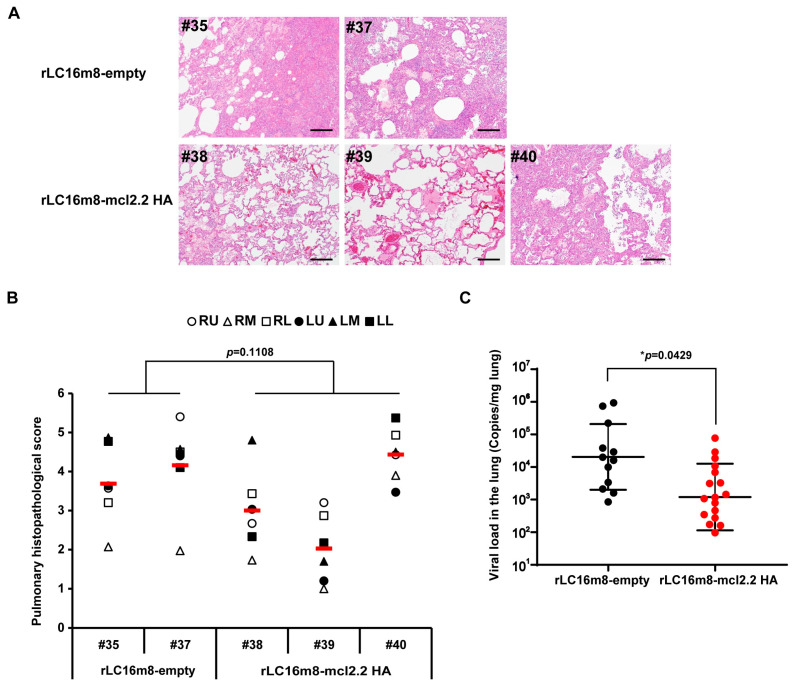
Histopathology and viral load in the lungs of cynomolgus macaques after HPAI H5N1 virus infection. (**A**) Representative lung sections (hematoxylin and eosin staining; section thickness 4 μm) at 7 days post-infection (dpi); original magnification was 100×. Bar, 200 μm. The number indicates animal ID. (**B**) Histopathological scores were obtained for each of the 15 defined regions of the lung lobe of each animal (RU, right upper; RM, right middle; RL, right lower; LU, left upper; LM, left middle; LL, left lower) 7 dpi with rLC16m8-empty or rLC16m8-mcl2.2 HA. Red horizontal bars indicate the mean pathological score in each group. *p* values were calculated using the Mann–Whitney U test. (**C**) Viral load in all lung lobes was determined by reverse transcription–quantitative PCR. The central horizontal value represents the geometric mean, and the whiskers indicate the geometric SD. *p* values were calculated by two-tailed non-paired Student’s *t*-tests. * *p* < 0.05.

**Figure 6 vaccines-13-00074-f006:**
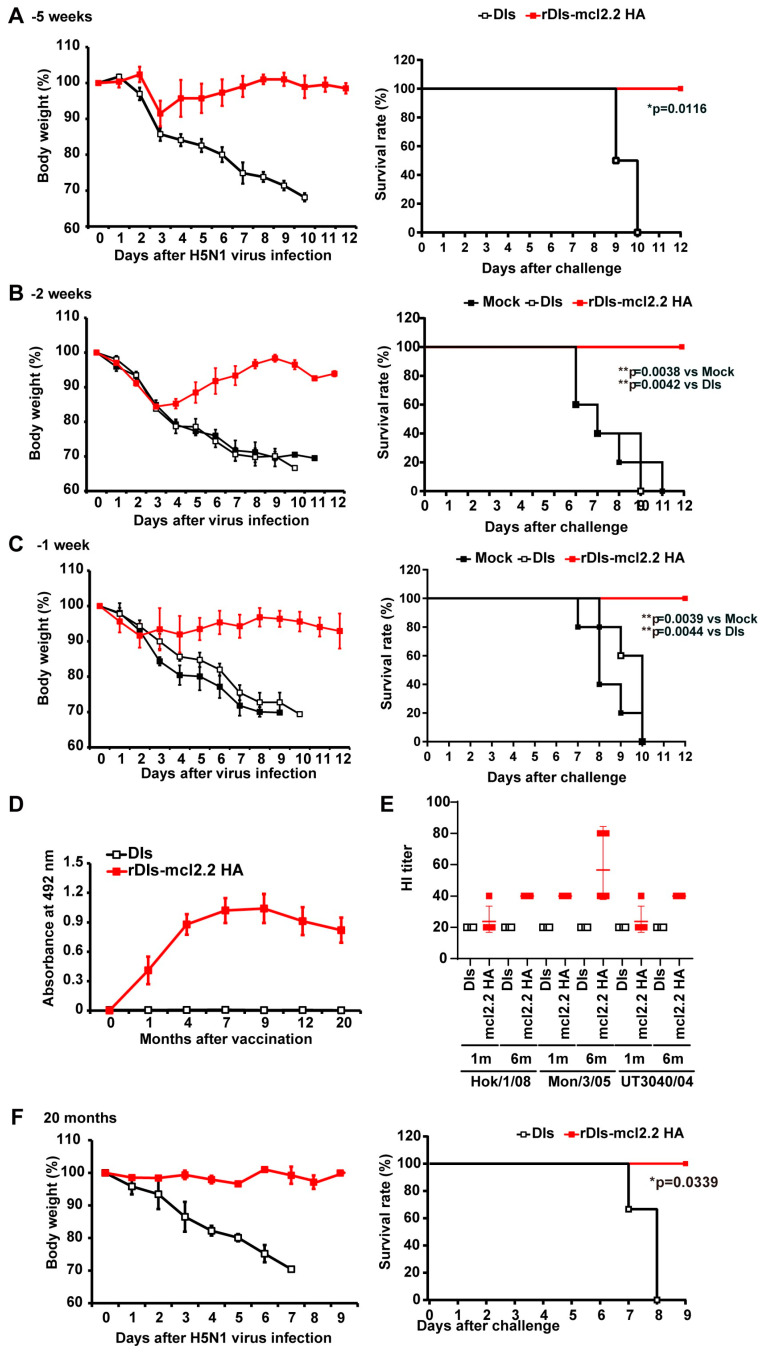
Rapid and long-term protection by a single dose of replication-deficient rDIs-mcl2.2 HA against lethal infection with H5N1 HPAI virus in mice. Female BALB/c mice were inoculated intradermally with 1 × 10^7^ PFU of rDIs-mcl2.2 HA or DIs. (**A**) Vaccinated mice (rDIs-mcl2.2 HA, *n* = 4; DIs, *n* = 4) were infected with 1 × 10^4^ PFUs of H5N1 A/whooper swan/Hokkaido/1/2008 (clade 2.3.2.1) 5 weeks after vaccination. Left: body weight was monitored daily after H5N1 virus infection. Right: survival rate was observed until 12 days post-infection (dpi). (**B**,**C**) The speed of protection by rDIs-mcl2.2 HA against the HPAI H5N1 virus was investigated. Two weeks (**B**) or one week (**C**) after vaccination, mice were infected intranasally with 166 × MLD_50_ of A/whooper swan/Hokkaido/1/2008 (clade 2.3.2.1) and then monitored daily for changes in their body weight (left panel) and survival rate until 12 dpi (right panel). (**D**,**E**) Long-term immunity by a single dose of rDIs-mcl2.2 HA was investigated. (**D**) The time course of production of IgG specific to the HA protein (clade 2.2) was measured by an ELISA. One thousand-fold diluted murine sera were used. (**E**) The HI titer of antisera from mice immunized with rDIs-mlc.2.2 HA (*n* = 4) or DIs (*n* = 4) 1 and 6 months after immunization. HI titers were determined against HPAI H5N1 A/whooper swan/Hokkaido/1/2008 (clade 2.3.2.1; Hok/1/08), A/whooper swan/Mongolia/3/2005 (clade 2.2; Mon/3/05), and A/Vietnam/UT3040/2004 (clade 1; UT3040/04) using 0.75% guinea pig erythrocytes. (**F**) Vaccinated mice were infected intranasally with 1 × 10^4^ PFUs of H5N1 A/whooper swan/Hokkaido/1/2008 (clade 2.3.2.1) after 20 months. Survival rates are compared in data shown in panels (**A**–**C**,**F**) using the Gehan–Breslow–Wilcoxon method. * *p* < 0.05, ** *p* < 0.01.

**Figure 7 vaccines-13-00074-f007:**
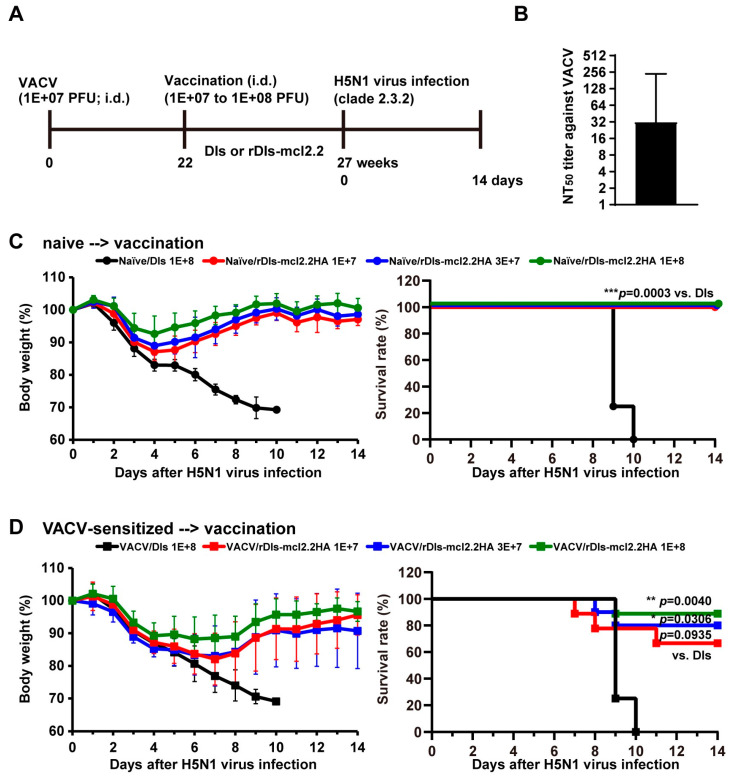
Protective efficacy of rDIs-mcl2.2 HA against the HPAI H5N1 virus in VACV-sensitized mice. (**A**) Experimental schedule. To investigate the protective efficacy of a single dose of rDIs-mcl2.2 HA against lethal infection with the HPAI H5N1 virus in VACV-sensitized mice, female BALB/c mice were sensitized intradermally with 1 × 10^7^ PFUs of the VACV LC16m8 strain and then immunized intradermally with rDIs-mcl2.2 HA (1 × 10^7^ PFUs, 3 × 10^7^ PFUs, or 1 × 10^8^ PFUs) 22 weeks after VACV sensitization (**D**). Age-matched naïve mice were used as controls (**C**). (**B**) A total of 21 weeks after VACV sensitization, the neutralization titer (50% neutralization) against LC16m8 was measured. Dashed lines denote the limits of detection. (**C**) Naïve mice were inoculated with rDIs-mcl2.2 HA and then infected intranasally with 166 × MLD_50_ of A/whooper swan/Hokkaido/1/2008 (clade 2.3.2.1) 5 weeks after vaccination. Body weight (left panel) and survival rate (right panel) were monitored daily. (**D**) VACV-sensitized mice were inoculated with rDIs-mcl2.2 HA and then infected intranasally with 166 × MLD_50_ of A/whooper swan/Hokkaido/1/2008 (clade 2.3.2.1) 5 weeks after vaccination. Body weight (left panel) and survival rate (right panel) were monitored daily. Survival rates are compared in panels (**C**,**D**) using the Gehan–Breslow–Wilcoxon method. * *p* < 0.05, ** *p* < 0.01, *** *p* < 0.001.

**Figure 8 vaccines-13-00074-f008:**
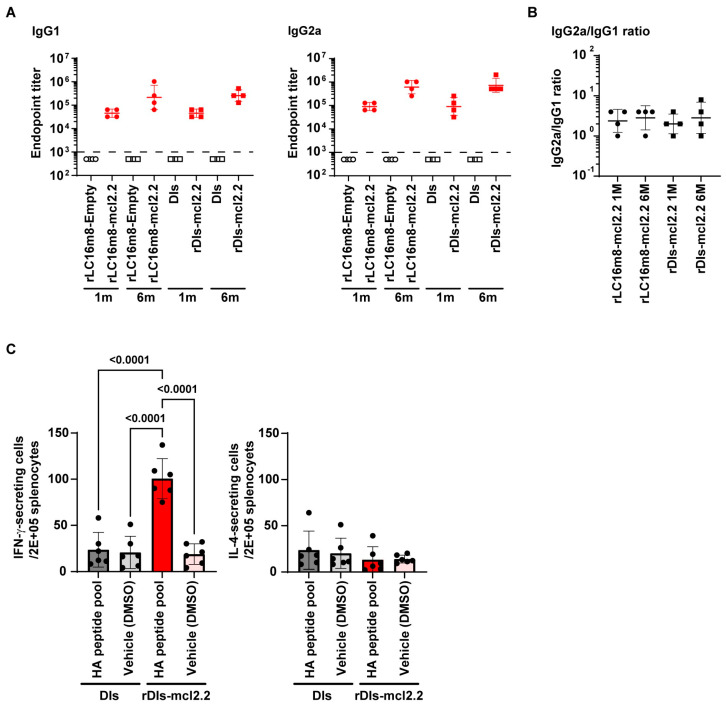
Th1/Th2 immune reponse to rVACV-mcl2.2 HA vaccine. (**A**) IgG1 (left) and IgG2a (right) responses against H5 HA clade 2.2 in mice vaccinated with rLC16m8-mcl2.2 HA (*n* = 4; red) or rDIs-mcl2.2 HA (*n* = 4) 1 and 6 months after immunization. Antisera from rLC16m8-empty-immunized (*n* = 4; white) and DIs-immunized (*n* = 4) mice were used as negative controls. The dashed line indicates a minimal dilution rate (1:1000) of antisera used in the ELISA. The endpoint titers of negative controls were defined as 500. (**B**) Th1/Th2 skewing responses in mice vaccinated with rLC16m8-mcl2.2 HA (*n* = 4) or rDIs-mcl2.2 HA (*n* = 4) 1 and 6 months after immunization. The IgG2a/IgG1 ratio was calculated using the respective endpoint titer values. (**C**) IFN-γ (left) or IL-4 (right) levels were measured via an ELISpot assay using the splenocytes of mice immunized with rDIs-mcl2.2 HA or DIs 1 month after immunization. Values are shown as the mean ± SD. Statistical analysis was performed using two-tailed one-way ANOVA with post hoc Tukey’s multiple comparison test.

## Data Availability

The original contributions presented in this study are included in the article/Appendix A. Further inquiries can be directed to the corresponding author.

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
