# Peer review of "Single Dose of Attenuated Vaccinia Viruses Expressing H5 Hemagglutinin Affords Rapid and Long-Term Protection Against Lethal Infection with Highly Pathogenic Avian Influenza A H5N1 Virus in Mice and Monkeys"

_vaccines, 2025, doi:10.3390/vaccines13010074_

Round 1
Reviewer 1 Report
Comments and Suggestions for Authors
In this paper entitled ‘Single dose of attenuated vaccinia viruses expressing H5 hemagglutinin affords rapid and long-term protection against lethal infection with highly pathogenic avian influenza A H5N1 virus in mice and monkeys” Yasui et al., described the protective efficiency of two recombinant vaccines-rLC16m8-mcl2.2 HA and rDIs-mcl2.2HA against HPIV (Highly pathogenic Influenza virus) in mice and monkey model. These are recombinant influenza vaccines against H5-subtypes generated by using two different strains of highly attenuated vaccinia virus vectors LC16m8 and rDIs.
The main findings of this study are below-
1. A single dose of rLC16m8-mcl2.2 HA showed protection in mice at both 1 week post vaccination and 20 months post vaccinations against heterologous HPAI H5N1 virus suggesting its broad cross-reactivity against various strains of given serotypes.
2. Interestingly a single dose of rLC16m8-mcl2.2 HA exhibited long-term protection from heterologous clad of HPAI H5N1 virus in cynomolgus macaques.
3. The DI strain, rDIs-mcl2.2HA though non replicative strain, also showed both rapid and long-lasting protection against HPA1 H5N1 infection suggesting the DIs-based vaccine might be safer for use in immunocompromised individuals.
4. Both vaccines also protected animals previously immunized with VCAV from challenges with the HPA1H5N1 virus. This demonstrates that the vaccine not only worked in naive animals but also in animals that already had pre-existing immunity to the VACV vector.
The manuscript is well-written, experimentally robust, and includes all the appropriate controls, adding weight to the current knowledge in the field of vaccines. I have a few suggestions to the authors to further enhance its quality and impact-
1. Line 296- ‘either’ spelling is wrong.
2. The study demonstrates protection and long-term immunity but does not study the underlying mechanisms behind this protection. For instance, analyzing which specific T-cell subtypes (CD4+, CD8+, or memory T cells), TH1/TH2 response could provide a clearer understanding of the effectiveness of these vaccines. Including these data can improve the impact of the manuscript.
3. While the study measured total IgG levels to see the effect of the vaccine on host humoral response, it did not assess specific IgG subtypes. IgG subtypes (such as IgG1, IgG2a, IgG2b, etc.) provide key information on the quality of the immune response, as each subtype has distinct roles in mediating immunity.
4. While the challenge study effectively demonstrated the vaccine's protective efficacy, it did not include adoptive transfer experiments to determine if antibodies generated by the vaccine could independently confer protection against these viruses. Adoptive transfer of immune serum to naive animals would help clarify whether antibody responses alone are sufficient for protection or if cell mediated immunity, play a key role.
Author Response
Response to reviewers
Reviewer #1
The authors thank the reviewers for their valuable suggestions. The comments were valuable and contributed significantly to improving the manuscript.
Point-by-point reply to Review #1
- Line 296- ‘either’ spelling is wrong.
As per your illustration, we have fixed the typo.
- The study demonstrates protection and long-term immunity but does not study the underlying mechanisms behind this protection. For instance, analyzing which specific T-cell subtypes (CD4+, CD8+, or memory T cells), TH1/TH2 response could provide a clearer understanding of the effectiveness of these vaccines. Including these data can improve the impact of the manuscript.
We thank the reviewer for the insightful comment regarding the protective mechanism of the rVACV-H5 HA vaccine. In future experiments, to elucidate these protective mechanisms, we plan to analyze the subtypes and phenotypes of antigen-specific T cells based on your suggestion. In this study, to investigate antigen-specific T-cell responses and the Th1/Th2 balance, we measured the production of IFN- and IL-4 in the splenocytes of BALB/c mice inoculated with rDIs-mcl2.2 using the ELISpot assay. We have added an explanation of these additional analyses to the Methods, Results, Discussion, and figure legends, including revisions in Figure 8C.
“ELISpot Assay
Four weeks after a single intradermal inoculation of rDIs-mcl2.2 HA into eight-week-old female BALB/c mice, splenocytes were harvested and stored in liquid nitrogen. Thawed splenocytes [2 ´ 105 cells/well] were seeded into the wells of 96-well MultiScreen IP Sterile plates (Merck Millipore Ltd, Burlington, MA) coated with anti-IFN-γ (MABTECH, Nacka strand, Sweden #3321-2H) or -IL-4 (MABTECH #3311-2H) antibody and subsequently stimulated with pooled H5 HA peptide (PepMix Influenza A (HA/Indonesia [H5N1]) [PM-INFA-HAIndo, JPT Peptide Technologies]) at 37 ºC for 48 h. The production of IFN-γ and IL-4 was analyzed using an ELISpot assay kit (MABTECH) according to the manufacturer’s instructions. After drying the ELISpot plates, the number of spots in each well was counted using an automated ELISpot plate reader (Advanced Imaging Devices GmbH, Strassberg, Germany).”
“ Th1-dominant responsiveness was also confirmed by antigen-specific T-cell responses in ELISpot assays. Mice vaccinated with rDIs-mcl2.2 HA showed significant production of IFN-γ upon antigen stimulation, whereas the IL-4 level was comparable to that in controls (Fig. 8C). Taken together, these results suggest that both rLC16m8-mcl2.2 HA and rDIs-mcl2.2 HA vaccines induce a Th1-dominated or balanced Th1/Th2 response.”
“We also demonstrated that vaccination with rVACV-H5 HA can induce both antigen-specific antibody production and T-cell responses that are Th1-dominant or Th1/Th2-balanced.”
- While the study measured total IgG levels to see the effect of the vaccine on host humoral response, it did not assess specific IgG subtypes. IgG subtypes (such as IgG1, IgG2a, IgG2b, etc.) provide key information on the quality of the immune response, as each subtype has distinct roles in mediating immunity.
Based on the reviewer’s constructive suggestion, we determined the endpoint titers of IgG1 and IgG2a specific to the HA protein using ELISA and analyzed the effect of Th1/Th2 balance on antigen-specific antibody responses by calculating the IgG2a/IgG1 ratio. We have added an explanation of these additional analyses to the Methods, Results, Discussion, and Figure Legends along with revisions in Figures 8A and B.
“The production of immunoglobulin G (IgG), IgG1, and IgG2a specific to H5 clade 2.2. HA protein in the sera of mice and IgG levels specific to the H5 HA protein in the sera of cynomolgus macaques was measured using ELISA.”
“To calculate the ratio of IgG2a/IgG1 specific to H5 HA in the sera of mice vaccinated with rDIs-mcl2.2 HA, endpoint titers of IgG2a and IgG1 specific for H5 HA were determined. The endpoint titer was defined as the reciprocal of the highest dilution of serum at which the absorbance at 490 nm exceeded two-fold the value of the blank.”
“Th1/Th2 immune response to influenza vaccine
Finally, we investigated the Th1/Th2 balance against rVACV-mcl2.2 HA because of the risk of worsening symptoms during subsequent pathogen infection associated with vaccine-induced immune responses, especially those with a Th1/Th2 balance skewed toward Th2. Antisera from mice immunized with rLC16m8-mcl2.2 HA or rDIs-mcl2.2 HA (1 and 6 months post-vaccination) were used to measure the endpoint titers of IgG1 and IgG2a specific to H5 HA using ELISA. The rLC16m8-mcl2.2 HA and rDIs-mcl2.2 HA vaccines potently induced both IgG1 and IgG2a (Fig. 8A), and the IgG2a/IgG1 ratio indicated a Th1-dominant phenotype after 1 and 6 months (Fig. 8B).”
- While the challenge study effectively demonstrated the vaccine's protective efficacy, it did not include adoptive transfer experiments to determine if antibodies generated by the vaccine could independently confer protection against these viruses. Adoptive transfer of immune serum to naive animals would help clarify whether antibody responses alone are sufficient for protection or if cell mediated immunity, play a key role.
We thank the reviewer for their insightful comments regarding the precise mechanisms by which the rVACV-H5 HA vaccine exerts broad protective effects against various H5N1 influenza viruses. We plan to conduct a more detailed analysis of the protective mechanism of the rVACV-H5 HA vaccine and publish our results in a new paper. Therefore, we have added the following sentences as a limitation in the Discussion section of this paper:
“In this study, we demonstrated that a single vaccination with rVACV-H5 HA could confer rapid and long-term protection against H5N1 viral infections in animals. We also demonstrated that vaccination with rVACV-H5 HA can induce both antigen-specific antibody production and T-cell responses that are Th1-dominant or Th1/Th2-balanced. However, the involvement of induced antibodies and cellular immunity in the protective effect of rVACV-H5 HA vaccination requires further study, such as passive transfer of antigen-specific antibodies, adoptive transfer of T cells, or T-cell depletion experiments.”
Reviewer 2 Report
Comments and Suggestions for Authors
The manuscript explores the efficacy of recombinant vaccinia virus-based vaccines encoding the H5 hemagglutinin (HA) protein for rapid and long-term protection against highly pathogenic avian influenza A (HPAI) H5N1 virus in mice and non-human primates. While the scientific findings are of interest for readers and community, the following should be considered for publication.
The method for pre-immunizing mice with VACV is absent. Please describe how pre-immunization with VACV was performed. Additionally, what method was used to validate VACV infection in pre-immunized mice? These results should be presented in the manuscript.
The manuscript lacks data on immune responses against the HA antigen in mice. Both humoral and cellular immune responses should be assessed and included, such as serum hemagglutinin inhibition titers and ELISpot data derived from splenic cells of immunized mice.
The statement describing clades 2.3.2.1 and 2.3.4 as homologous is incorrect. These clades are not homologous but closely related. Please revise the manuscript throughout accordingly.
Author Response
Response to reviewers
Reviewer #2
The authors thank the reviewers for their valuable suggestions. The comments were appreciated and aided significantly in improving the manuscript.
Point-by-point reply to Review #2
The method for pre-immunizing mice with VACV is absent. Please describe how pre-immunization with VACV was performed. Additionally, what method was used to validate VACV infection in pre-immunized mice? These results should be presented in the manuscript.
We not providing a depiction of the method used for pre-immunizing mice with VACV. According to your suggestion, we have added the following sentences to the Methods section:
“In the pre-immunization VACV study, BALB/c mice were inoculated intradermally with 1 × 107 PFUs of LC16m8 or rLC16m8-empty. After 4 weeks, the rLC16m8-empty-immunized mice were intradermally inoculated with either 1 ´ 107 PFU of rLC16m8-mcl2.2 HA or rLC16m8-empty and inoculated intranasally with 1 ´ 104 PFU of the H5N1 A/whooper swan/Hokkaido/1/2008 strain. In contrast, sera were collected 21 weeks after LC16m8 inoculation from LC16m8-immunized mice and used to determine the neutralization titer against LC16m8 using the plaque assay, as described below. LC16m8-immunized mice were divided into four subgroups, in which the neutralization titer against VACV was similar within each subgroup (Supplementary Fig. 5), and inoculated intradermally with rDIs-mcl2.2 HA (1 × 107, 3 × 107, or 1 × 108 PFU/mouse) or DIs (1 × 108 PFU/mouse). All mice were infected with 1 ´ × 104 PFUs of the H5N1 A/whooper swan/Hokkaido/1/2008 strain 5 weeks after vaccination.”
The manuscript lacks data on immune responses against the HA antigen in mice. Both humoral and cellular immune responses should be assessed and included, such as serum hemagglutinin inhibition titers and ELISpot data derived from splenic cells of immunized mice.
We thank the reviewer for the insightful comment regarding the protective mechanism of the rVACV-H5 HA vaccine. Based on your suggestion, we analyzed the HI titer of the sera of BALB/c mice vaccinated with rLC16m8-mcl2.2 HA or rDIs-mcl2.2 HA against three different clades of H5N1 HPAI viruses (A/whooper swan/Hokkaido/1/2008 [Clade 2.3.2.1] and A/whooper swan/Mongolia/3/2005 [clade 2.2] strains, and A/Vietnam/UT3040/2004 [clade 1]). Furthermore, we measured the production of IFN- and IL-4 levels in the splenocytes of BALB/c mice inoculated with rDIs-mcl2.2 using the ELISpot assay. We have added an explanation of these additional analyses to the Methods, Results, Discussion, and Figure Legends along with revisions to Figures 3B, 6E, and 8C.
“Hemagglutination inhibition (HI) assay
HI assays were performed according to standard methods [25]. Briefly, serum samples were pretreated (37 °C, 18 h) with a receptor-destroying enzyme (RDE II, Denka Seiken, Tokyo, Japan) and then inactivated using heat (56 °C, 30 min). The resulting serum samples were subjected to serial two-fold dilutions with PBS and then mixed with a 4 × hemagglutination (HA) titer of H5N1 A/whooper swan/Hokkaido/1/2008, A/whooper swan/Mongolia/3/2005, or A/Vietnam/UT3040/ 2004 in a 0.75% suspension of guinea pig erythrocytes. After 1 h of incubation at room temperature, hemagglutination was determined by visual inspection. Titers were expressed as the reciprocal of the maximum dilution of serum that completely inhibited hemagglutination.”
“ELISpot Assay
Four weeks after a single intradermal inoculation of rDIs-mcl2.2 HA into eight-week-old female BALB/c mice, splenocytes were harvested and stored in liquid nitrogen. Thawed splenocytes [2 ´ 105 cells/well] were seeded into the wells of 96-well MultiScreen IP Sterile plates (Merck Millipore Ltd, Burlington, MA) coated with anti-IFN-γ (MABTECH, Nacka strand, Sweden #3321-2H) or -IL-4 (MABTECH #3311-2H) antibody and subsequently stimulated with pooled H5 HA peptide (PepMix Influenza A (HA/Indonesia [H5N1]) [PM-INFA-HAIndo, JPT Peptide Technologies]) at 37 ºC for 48 h. The production of IFN-γ and IL-4 was analyzed using an ELISpot assay kit (MABTECH) according to the manufacturer’s instructions. After drying the ELISpot plates, the number of spots in each well was counted using an automated ELISpot plate reader (Advanced Imaging Devices GmbH, Strassberg, Germany).”
“HI activities against three different clades of the H5N1 virus in the sera of rLC16m8-mcl2.2-immunized mice were also higher 6 months after vaccination than those recorded 1 month after vaccination (Fig. 3B).”
“HI activities against three different clades of the H5N1 virus in the sera of rDIs-mcl2.2-immunized mice were higher 6 months after vaccination than those recorded 1 month after vaccination (Fig. 6E), similar to those observed with rLC16m8-mcl2.2 (Fig. 3B).”
“Th1-dominant response was also confirmed by antigen-specific T-cell responses in ELISpot assays. Mice vaccinated with rDIs-mcl2.2 HA showed significant production of IFN-γ upon antigen stimulation, whereas the IL-4 level was comparable to that in controls (Fig. 8C). Taken together, these results suggest that both rLC16m8-mcl2.2 HA and rDIs-mcl2.2 HA vaccines induce a Th1-dominant or balanced Th1/Th2 response.”
“We also demonstrated that vaccination with rVACV-H5 HA can induce both antigen-specific antibody production and T-cell responses that are Th1-dominant or Th1/Th2-balanced.”
The statement describing clades 2.3.2.1 and 2.3.4 as homologous is incorrect. These clades are not homologous but closely related. Please revise the manuscript throughout accordingly.
In accordance with the reviewer’s comment, we have revised the manuscript to state that clades 2.3.2.1 and 2.3.4 are closely related.